# Cholinergic Receptor Nicotinic Beta 2 Subunit Promotes the Peritoneal Disseminating Metastasis of Colorectal Cancer

**DOI:** 10.3390/cancers17152485

**Published:** 2025-07-28

**Authors:** Shinichi Umeda, Kenshiro Tanaka, Takayoshi Kishida, Norifumi Hattori, Haruyoshi Tanaka, Dai Shimizu, Hideki Takami, Masamichi Hayashi, Chie Tanaka, Goro Nakayama, Mitsuro Kanda

**Affiliations:** Department of Gastrointestinal Surgery, Nagoya University Graduate School of Medicine, 65 Tsurumai-cho, Showa-ku, Nagoya 466-8550, Japan; tanaka.kenshiro.i8@s.mail.nagoya-u.ac.jp (K.T.); kishida.takayoshi.v1@f.mail.nagoya-u.ac.jp (T.K.); hattori.norifumi.v1@f.mail.nagoya-u.ac.jp (N.H.); tanaka.haruyoshi.f1@f.mail.nagoya-u.ac.jp (H.T.); shimizu.dai.t2@f.mail.nagoya-u.ac.jp (D.S.); takami.hideki.r7@f.mail.nagoya-u.ac.jp (H.T.); hayashi.masamichi.b9@f.mail.nagoya-u.ac.jp (M.H.); tanaka.chie.u9@f.mail.nagoya-u.ac.jp (C.T.); nakayama.goro.z5@f.mail.nagoya-u.ac.jp (G.N.); kanda.mitsuro.v5@f.mail.nagoya-u.ac.jp (M.K.)

**Keywords:** colorectal cancer, cholinergic receptor nicotinic beta 2 subunit, peritoneal recurrence, biomarker, expression

## Abstract

Colorectal cancer with peritoneal metastasis is associated with a particularly poor prognosis, underscoring the urgent need for novel therapeutic strategies. In this study, CHRNB2 was investigated as a potential molecular target. Knockdown of CHRNB2 in colorectal cancer cell lines significantly suppressed cell proliferation, migration, and invasion and notably reduced the ability to form peritoneal metastases. Furthermore, treatment with anti-CHRNB2 monoclonal antibodies inhibited cell proliferation, suggesting potential therapeutic utility. Clinical analysis of colorectal cancer specimens revealed a significant correlation between high CHRNB2 expression and increased risk of peritoneal recurrence, indicating its involvement in metastatic progression. These findings suggest that CHRNB2 plays a critical role in the malignant phenotype associated with peritoneal dissemination and may serve as a promising target for therapeutic intervention in colorectal cancer patients with or at risk for peritoneal metastasis.

## 1. Introduction

The prognosis of advanced or recurrent colorectal cancer (CRC) has improved with the development of preoperative and postoperative chemotherapy [1,2,3], radiation therapy [4], and advances in surgical techniques, enabling curative resection of liver and lung metastases [5,6]. On the other hand, curative resection of peritoneal dissemination in CRC remains challenging [7], and anticancer drugs are less effective in this context [8]; therefore, the prognosis is poorer than that for liver or lung metastases. There have been several reports on cytoreductive surgery (CRS) combined with hyperthermic intraperitoneal chemotherapy (HIPEC) for peritoneal dissemination of CRC [9,10,11], and recent meta-analyses on peritoneal metastasis have reported the potential survival benefit of CRS plus HIPEC [12], as well as the effectiveness of postoperative chemotherapy following CRS plus HIPEC [13]. These findings are also referenced in the NCCN guidelines. Nevertheless, the availability of this approach is limited to specialized centers, and it has not yet been established as a standard treatment. Furthermore, the sensitivity of computed tomography (CT) and magnetic resonance imaging (MRI) for detecting peritoneal dissemination is limited, making early detection and treatment initiation difficult [14]. Accordingly, there is an urgent need to identify diagnostic biomarkers and therapeutic target genes specific to peritoneal dissemination of CRC to improve clinical outcomes. Our group previously conducted a comprehensive gene expression analysis of biological samples from patients with metastatic gastric cancer and identified aberrantly high expression of the cholinergic receptor nicotinic beta 2 subunit (*CHRNB2*) in cases with distant metastases [15]. To explore the applicability of this finding in CRC, preliminary experiments using clinical CRC specimens were conducted, which confirmed a correlation between CHRNB2 expression and peritoneal dissemination. CHRNB2 is a member of the nicotinic acetylcholine receptor family, a transmembrane protein complex composed of five subunits surrounding a central ion channel [16]. CHRNB2 has been reported to be involved in familial epileptic seizures [17], Alzheimer’s disease [18], and nicotine dependence related to nicotine metabolism [19,20]. In addition, its involvement in pancreatic cancer and gastric cancer has also been reported [21], but there have been no reports on CRC. In this study, CHRNB2 function in CRC cell lines was analyzed, and the association between CHRNB2 expression and clinicopathological factors in clinical colorectal cancer was investigated to demonstrate whether it can be a diagnostic biomarker and therapeutic target for peritoneal dissemination of CRC.

## 2. Materials and Methods

### 2.1. Ethics

This study was carried out in accordance with the ethical principles outlined in the Declaration of Helsinki by the World Medical Association, which governs research involving human participants. Approval was obtained from the Institutional Review Board of Nagoya University Hospital (approval no. 2014-0116). In line with the Board’s guidelines, written informed consent was obtained from all patients for the use of their clinical data and biological specimens.

### 2.2. Cell Lines Collection

The CRC cell lines LOVO, CaR-1, DLD-1, SW1116, HT-29, COLO 201, SW1417, SW1083, and RKO were obtained from the American Type Culture Collection (Manassas, VA, USA).

### 2.3. Quantitative Reverse-Transcription Polymerase Chain Reaction (qRT-PCR)

Total RNA (10 μg per sample) was extracted from both cells and tissue samples using the RNeasy Mini Kit (Qiagen, Hilden, Germany). Complementary DNA (cDNA) was synthesized and subjected to PCR amplification with gene-specific primers listed in Appendix A. Real-time quantification was conducted by monitoring SYBR^®^ Green fluorescence using the ABI StepOnePlus Real-Time PCR System (Applied Biosystems, Foster City, CA, USA). The expression level of glyceraldehyde-3-phosphate dehydrogenase (GAPDH) served as an internal control, as reported previously [22].

### 2.4. shRNA Mediated CHRNB2 Knock-Down in CRC Cell Lines

CHRNB2 knockdown (KD) was performed using the MISSION TRC shRNA Lentiviral Particles Target Set, and negative control was performed using Negative Control shRNA Lentiviral Particles pLKO.1-puro Control (Thermo Fishier SCIENCE, Waltham, MA, USA). A total of 1.6 × 10^4^ cells were seeded in 96-well plates and incubated at 37 °C for 18 h, after which hexadimethrine bromide (final concentration 8 μg/mL) and 5 μL of lentiviral particles were added. The cells were then incubated at 37 °C for 20 h with a medium containing puromycin. Then, the cell line was incubated with a new medium containing puromycin, replacing the medium every 3 days until resistant colonies emerged. Five puromycin-resistant colonies were picked up, KD efficiency was measured, and two efficient sh-CHRNB2-treated cell lines were selected.

### 2.5. Generation of Anti-CHRNB2 Antibodies

Polyclonal anti-CHRNB2 antibodies were generated by immunizing rabbits with synthetic peptides containing CHRNB2 epitopes, which were designed based on in silico immunogenicity predictions. Peptides were synthesized using a solid-phase synthesis method. For the generation of anti-CHRNB2 monoclonal antibodies (mAb), three six-week-old female BALB/c mice were immunized twice at three-week intervals with 40 μg of the peptide. The mouse exhibiting the highest antibody titer, as measured by enzyme-linked immunosorbent assay (ELISA), was subsequently boosted with 40 μg of peptide. Hybridoma cells were then generated, and ELISA-positive clones were isolated by limiting dilution. The supernatants containing antibodies were evaluated for their ability to inhibit the proliferation of CRC cells. Detailed methodologies are described in previously published reports. The tumor-suppressive effects of the anti-CHRNB2 mAb were compared among untreated cells, cells treated with 10% control IgG, and cells treated with 10% CHRNB2 mAb [15].

### 2.6. Assays of Cell Proliferation, Migration, and Invasion

Cell proliferation was assessed using the Cell Counting Kit-8 (Dojindo Molecular Technologies, Inc., Kumamoto, Japan). Cells (5 × 10^3^ per well) were cultured, and cell viability was determined by measuring the absorbance in each well on days 1, 3, and 5 following the addition of 10 µL of the CCK-8 reagent. The invasive capacity of CRC cells through Matrigel was analyzed using BioCoat Matrigel invasion chambers (BD Biosciences, Bedford, MA, USA) in accordance with the manufacturer’s instructions. Cells (2.5 × 10^4^) suspended in serum-free RPMI medium were seeded into the upper chamber. After 48 h of incubation, cells that had migrated to the lower surface of the membrane were fixed, stained, and counted in eight randomly chosen microscopic fields under 200× magnification. Cell migration was further examined using a wound healing assay, and the width of the wound area was measured at 40× magnification as previously described [22].

### 2.7. Mouse Subcutaneous Xenograft Model

The Animal Research Committee of Nagoya University approved all animal experiments. Tumor burden did not exceed the recommended dimensions, and the animals were anesthetized and sacrificed using acceptable methods. A total of 1 × 10^6^ cells resuspended in 50 µL of PBS were injected into the bilateral backs of 6-week-old male BALB/c (nu/nu) mice (*n* = 4) (SLC, Inc., Hamamatsu, Japan). Tumors were collected 8 weeks after implantation.

### 2.8. Mouse Peritoneal Dissemination Model

A total of 2 × 10^6^ cells were resuspended in 1 mL of PBS and injected into the peritoneal cavity of 6-week-old male BALB/c (nu/nu) mice (*n* = 3) (SLC, Inc., Hamamatsu, Japan). Mice were sacrificed 6 weeks after implantation.

### 2.9. Analysis of Clinical Samples

Three hundred one CRC tissues were obtained from patients who underwent curative resection in our institution between 2007 and 2017. Correlations between CHRNB2 mRNA levels and clinicopathological findings and prognosis were evaluated. A freely available integrated dataset (*n* = 1002 CRC patients) was accessed at http://kmplot.com/analysis/ (accessed on 15 May 2025).

### 2.10. Statistical Analysis

Statistical differences between two groups in terms of average values were assessed using the Mann–Whitney U test, while comparisons of proportions were carried out using the χ^2^ test. Spearman’s rank correlation coefficient was utilized to evaluate pairwise correlations between variables. The Kaplan–Meier method was used to estimate cumulative recurrence rates, with differences between groups evaluated by the log-rank test. Cox proportional hazards models were applied for both univariable and multivariable analyses. A *p*-value less than 0.05 was considered statistically significant for all tests. Statistical analyses were conducted using JMP version 17 software (SAS Institute, Inc., Cary, NC, USA).

## 3. Results

### 3.1. CHRNB2 mRNA Expression in CRC Cell Lines and CHRNB2 KD Efficiency

CHRNB2 mRNA expression was sparse in nine cell lines and particularly high in LOVO, followed by CaR1, DLD1, and SW1116 (Figure 1A). Based on these results, LOVO was selected for CHRNB2 KD. Two kinds of KD cell lines (sh-CHRNB2-1, sh-CHRNB2-2) and sh-negative with no target shRNA were established. KD efficiency of sh-CHRNB2-1 and sh-CHRNB2-2 was 80% and 67%, respectively (Figure 1B).

### 3.2. Cell Function Analysis

Analysis of proliferative ability revealed that sh-CHRNB2-1 cells showed a significant decrease in proliferative ability on Day 3 and Day 5 after seeding, and sh-CHRNB2-2 cells showed a significant decrease on Day 5 compared to the wild-type and sh-negative cells (Figure 1C). Analysis of invasive capacity showed that Matrigel-infiltrating cells were reduced by 46% in sh-CHRNB2-1 and 69% in sh-CHRNB2-2 (Figure 1D). Migration assay showed a significant decrease in migration ability of sh-CHRNB2-2 cells at 48 h after the start of migration, while CHRNB2-1 showed a slight decrease in migration capacity, but the difference was not significant (Figure 2A).

### 3.3. Effect of the CHRNB2 mAb

Our results demonstrated that the mAb significantly reduced cell proliferation on Days 3 and 5 in LOVO cells, and on Days 5 and 7 in CaR1 cells, compared with both control IgG-treated and untreated groups (Figure 2B).

### 3.4. Mouse Subcutaneous Tumor Model

Tumor growth at 8 weeks after subcutaneous tumor injection was not different between WT and negative control, with an average tumor diameter of 10 mm. On the other hand, sh-CHRNB21-1 and sh-CHRNB2-2 showed significantly reduced growth compared to WT and negative control. In particular, sh-CHRNB21-1 showed strong growth inhibition (Figure 3A).

### 3.5. Mouse Peritoneal Dissemination Model

One death of a mouse with weight loss was observed in each of the wild-type and sh-negative groups. One mouse in the wild-type and two mice in the sh-negative group showed peritoneal dissemination at 6 weeks after seeding. On the other hand, none of the three mice in the sh-CHRNB2-1 and sh-CHRNB21-2 groups formed peritoneal dissemination (Figure 3B).

### 3.6. CHRNB2 mRNA Expression Analysis of Clinical Samples

The analysis of 301 resected specimens from stage II–III CRC patients, divided into two groups, high (*n* = 75) and low (*n* = 226) CHRNB2 expression groups, using the third quartile of CHRNB2 expression at the cancer site as the cutoff value, showed a trend toward slightly deeper tumor depth and more lymphatic involvement in the high CHRNB2 group; however, there was no significant difference. The third quartile was selected as the cutoff value, as it closely approximated the optimal threshold identified by ROC analysis. There were no significant differences in age, gender, tumor site, tumor markers, and pathological stage between the two groups (Appendix A).

### 3.7. Kaplan–Meier Analysis

Analysis of disease-free survival (DFS) and overall survival (OS) showed a slightly worse prognosis in the high CHRNB2 group (HR 1.51, 95% CI 0.90–2.53, *p* = 0.120; HR 2.02 95% CI 0.90–4.52, *p* = 0.085, respectively), but not significantly different (Figure 4A,B). Cumulative recurrence of peritoneal dissemination was significantly higher in the high CHRNB2 group (HR 3.29, 95% CI 1.23–8.79, *p* = 0.017) (Figure 5A). On the other hand, cumulative liver recurrence (HR 1.10, 95% CI 0.36–3.42, *p* = 0.865), cumulative lung recurrence (HR 1.23, 95% CI 0.58–2.80, *p* = 0.607), and cumulative nodal recurrence (HR 1.34, 95% CI 0.42–4.28, *p* = 0.619) were not significantly different in the high and low CHRNB2 groups (Figure 5B–D). Kaplan–Meier plotter analysis showed a significantly worse prognosis for OS and DFS in the high CHRNB2 group (HR 1.45, 95% CI 1.1–1.92, *p* = 0.0078; HR 2.02, 95% CI 1.71–2.52, *p* = 0.0064, respectively) (Appendix A).

### 3.8. Risk Factor for Peritoneal Recurrence

In univariable analysis, tumor size (≥40 mm) (*p* = 0.0293, HR 4.03, 95% CI 1.15–14.2), high CEA (*p* = 0.0167, HR 3.44, 95% CI 1.25–9.49), tumor depth (T4) (*p* < 0.0001, HR 9.27, 95% CI 3.36–25.6), and high CHRNB2 (*p* = 0.0173, HR 3.29, 95% CI 1.23–8.79) were prognostic factors for peritoneal recurrence. On multivariable analysis, tumor depth (T4) (*p* < 0.0041, HR 5.69, 95% CI 1.95–16.6) and high CHRNB2 (*p* = 0.0417, HR 2.83, 95% CI 1.03–16.6) were independent prognostic factors for peritoneal recurrence (Table 1).

## 4. Discussion

In the present study, we demonstrated that *CHRNB2* is associated with malignant phenotypes in CRC, particularly with peritoneal recurrence, based on analyses of CRC cell lines and clinical specimens. KD of CHRNB2 reduced proliferative capacity and also suppressed migration and invasion functions essential for metastatic progression [23], suggesting that CHRNB2 contributes to CRC metastasis.

Interestingly, while CHRNB2 KD cells exhibited partial tumor growth in proliferation and subcutaneous tumor models, they failed to form peritoneal metastases in a dissemination model. This suggests that CHRNB2 may play a critical role in the malignant phenotype of CRC, particularly in peritoneal dissemination, and could represent a potential therapeutic target for preventing peritoneal metastasis.

Our research group has previously succeeded in generating an anti-CHRNB2 mAb, which has shown growth-inhibitory effects on gastric cancer cell lines following intraperitoneal administration [15]. In this experiment, the growth-inhibitory effect of CHRNB2 mAb on colon cancer cell lines was demonstrated. Therefore, CHRNB2 may be a useful option for colon cancer peritoneal metastasis treatment in the future.

As a mechanism by which CHRNB2 inhibits the metastasis of CRC, CHRNB2 has been reported to mediate cell survival, resistance to anticancer drugs, and stemness of cancer cells by inhibiting the activity of components of the PI3K–AKT and JAK–STAT signaling pathways [15]. Another study reported that CHRNB2 regulates EMT by modulating the Wnt/β-catenin pathway and that CHRNB2 is involved in EMT by directly regulating the expression of several genes such as SRY, SOX17, and SOX6 [21]. In colorectal cancer, the PI3K–AKT–mTOR signaling axis plays a central role in regulating cell proliferation, survival, and metabolism, and is activated in approximately 40% of cases [24]. Additionally, meta-analyses have reported that β-catenin is associated with colorectal cancer progression [25]. Based on these findings, it is suggested that CHRNB2 may regulate the malignant phenotype of colorectal cancer by modulating downstream signaling cascades, including the PI3K–AKT–mTOR and β-catenin pathways.

Analysis of clinical specimens showed that OS and DFS tended to be worse in the high CHRNB2 group, but there were no significant differences. However, analysis of cumulative recurrence showed that liver, lymph node, and lung recurrence did not differ significantly by CHRNB2 value, while cumulative peritoneal recurrence was significantly higher in the high CHRNB2 group, and almost no peritoneal recurrence was observed in the low CHRNB2 group. This is consistent with the result that KD cells that proliferated to some extent in the subcutaneous tumor model did not form peritoneal dissemination at all. These results suggest that CHRNB2 may play an essential role in the formation of peritoneal dissemination of CRC cells. Analysis of clinicopathological factors and CHRNB2 expression showed no correlation with tumor markers, tumor depth, and lymph node metastasis, and high CHRNB2 was an independent poor prognostic factor along with wall depth T4 in multivariate analysis. This suggests that not only tumor factors that worsen over time, but also the malignant potential acquired by the tumor itself, namely high CHRNB2 expression, are important for the formation of peritoneal dissemination.

CHRNB2 may also have clinical applications. First, patients with high CHRNB2 levels should be observed closely during surgery for peritoneal dissemination or undergo ascites cytology to detect dissemination at an early stage [26], and, if possible, resection should be performed [27]. Second, peritoneal dissemination is difficult to diagnose by imaging; however, CHRNB2 measurement in serum may be a useful marker for recurrence of peritoneal dissemination [28]. Third, CHRNB2 expression in the primary tumor may also be used as a companion diagnosis when selecting anti-CHRNB2 antibody therapy in the future.

This study has several limitations. First, the retrospective design and small cohort size limit the generalizability of the findings; prospective validation in larger cohorts is needed. Second, the analysis of CHRNB2 protein expression has not been performed, and the precise molecular mechanism underlying CHRNB2-mediated peritoneal dissemination remains unclear. Third, further in vivo studies using intraperitoneal and intravenous administration of anti-CHRNB2 antibodies are required to assess their potential for clinical translation. Fourth, this study involved an older patient cohort, and comprehensive molecular data such as RAS mutation status and MSI were not available. Therefore, consensus molecular subtypes classification could not be evaluated.

## 5. Conclusions

In conclusion, CHRNB2 contributes to the malignant phenotype of CRC, especially in promoting peritoneal dissemination. These findings suggest that CHRNB2 may serve as a potential diagnostic biomarker and therapeutic target for peritoneal metastasis in CRC.

## Figures and Tables

**Figure 1 cancers-17-02485-f001:**
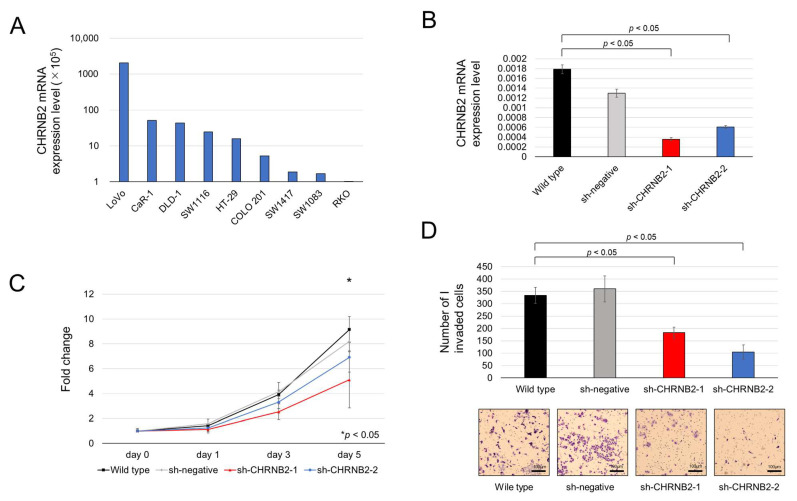
(**A**) CHRNB2 mRNA expression in nine CRC cell lines. (**B**) shRNA mediated knockdown efficacy in LOVO cells. (**C**) Proliferation of wild-type, sh-negative, and two kinds of sh-CHRNB2 cells. The fold changes from Day 0 are shown (* *p* < 0.05). (**D**) Invasion assays of wild-type, sh-negative, and two kinds of sh-CHRNB2 cells. The graph shows the number of cells that invaded the Matrigel chamber per section.

**Figure 2 cancers-17-02485-f002:**
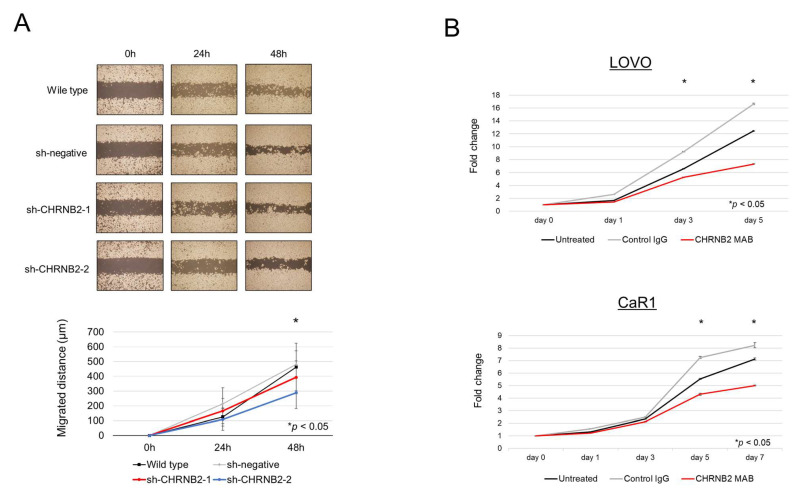
(**A**) Images (upper) and quantification (lower) of the results of migration assays of wild-type, sh-negative, and two kinds of sh-CHRNB2 cells. (**B**) Effect of CHRNB2 monoclonal antibody on the proliferation of LOVO and CaR1 cells.

**Figure 3 cancers-17-02485-f003:**
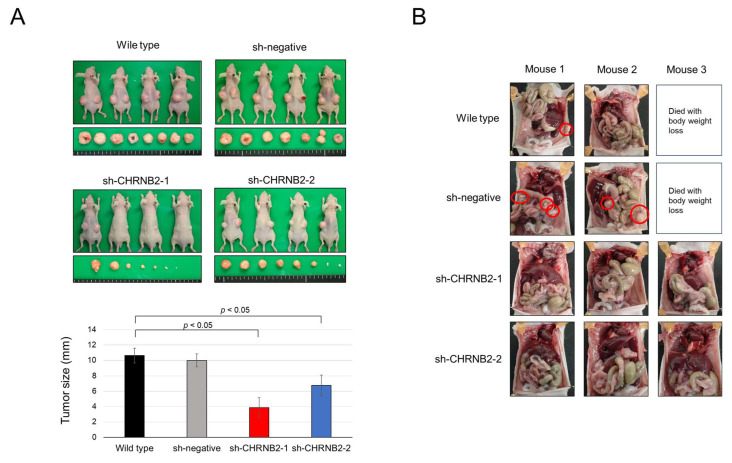
(**A**) Percutaneous tumor model. Images of mice sacrificed 8 weeks after implantation and excised tumors (upper), and quantification of tumor size 8 weeks after subcutaneous injection (lower), are shown. (**B**) Peritoneal dissemination model. Laparotomy findings 6 weeks after dissemination are shown. Dissemination nodules are indicated by red circles.

**Figure 4 cancers-17-02485-f004:**
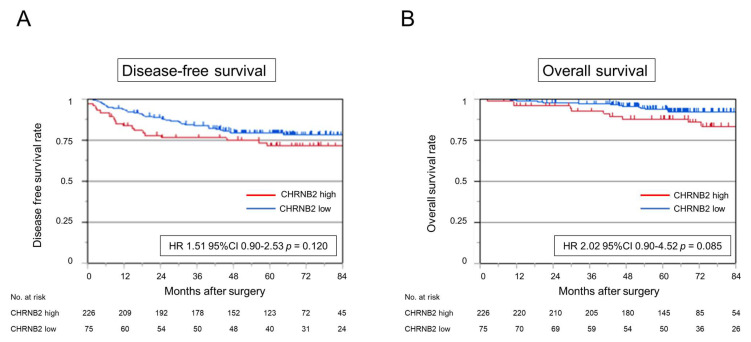
Kaplan–Meier analysis of patients with stage II–III CRC. Patients were divided into high and low CHRNB2 expression groups using the third quartile of CHRNB2 expression at the cancer site. (**A**) Disease-free survival rate. (**B**) Overall survival rate.

**Figure 5 cancers-17-02485-f005:**
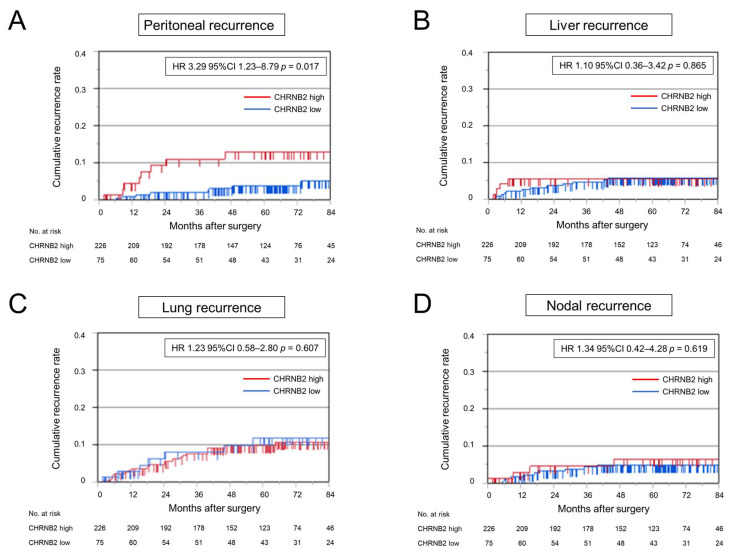
Kaplan–Meier analysis of cumulative recurrence rate in high and low CHRNB2 expression groups. (**A**) Cumulative peritoneal recurrence. (**B**) Cumulative liver recurrence. (**C**) Cumulative lung recurrence. (**D**) Cumulative nodal recurrence.

**Table 1 cancers-17-02485-t001:** Prognostic factors for peritoneal recurrence of 301 stage II and III resectable colorectal cancer patients.

	Univariable			Multivariable		
	Hazard Ratio	95% CI	*p*	Hazard Ratio	95% CI	*p*
Age (≥65)	0.92	0.33–2.55	0.883			
Gender (male)	2.29	0.83–6.89	0.105			
Tumor location (right)	2.19	0.82–5.84	0.116			
Tumor size (≥40 mm)	4.03	1.15–14.2	0.029 *	2.25	0.60–8.48	0.226
Carcinoembryonic antigen (>5 ng/mL)	3.44	1.25–9.49	0.016 *	2.30	0.81–6.50	0.115
Carbohydrate antigen 19-9 (>37 IU/mL)	2.13	0.77–5.87	0.142			
Approach (open)	5.01	1.87–13.3	0.001 *	1.90	0.66–5.54	0.234
Lymph node dissection (<D3)	2.65	0.96–7.30	0.059			
Tumor depth (pT4)	9.27	3.36–25.6	<0.001 *	5.69	1.95–16.6	0.001 *
Lymph node metastasis	1.14	0.43–3.05	0.783			
Tumor differentiation (undifferentiated)	0.96	0.13–7.29	0.970			
Lymphatic involvement (ly+)	1.63	0.46–5.73	0.444			
Vascular invasion (v+)	2.86	0.81–10.0	0.100			
Postoperative adjuvant chemotherapy	1.21	0.43–3.32	0.713			
High CHRNB2 expression	3.29	1.23–8.79	0.017 *	2.83	1.03–16.6	0.041 *

* Statistically significant in multivariable analysis. CI, confidence interval; CHRNB2, cholinergic receptor nicotinic beta 2 subunit.

## Data Availability

The data presented in this study are available upon request from the corresponding author. The data are not publicly available due to privacy concerns involving sensitive patient information and the ongoing nature of the clinical study. Data sharing is limited to safeguard participant confidentiality and is in compliance with institutional ethical guidelines.

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
