# Peer review of "Cholinergic Receptor Nicotinic Beta 2 Subunit Promotes the Peritoneal Disseminating Metastasis of Colorectal Cancer"

_cancers, 2025, doi:10.3390/cancers17152485_

Round 1
Reviewer 1 Report
Comments and Suggestions for Authors
In this manuscript Umeda et al. have investigated the role of the cholinergic receptor nicotinic beta 2 subunit CHRNB2 in colorectal cancer (CRC) metastasis, with a specific focus on peritoneal spread.
They utilized cell lines, performed knockdown and xenograft experiments. Additionally, they explored institutional patient’s specifically 301 stage II–III CRC patients and performed univariate and multivariate analysis.
The authors have concluded that CHRNB2 plays a role in peritoneal metastasis and may be a biomarker and therapeutic target with significant promise.
This manuscript addresses an important and critical aspect of CRC metastasis. The multi-pronged analysis of in vitro, in vivo, and institutional clinical cohort is a strength, but key mechanistic experiments, and expanded discussion in addition to statistical rigor are needed before publication.
Below are my observations:
- The authors need to strengthen the downstream signalling specifically in context of CRC.
- The patient cohort spans 2007–2017; please add another table for discussing clinico-pathological features/characteristics in detail.
- The manuscript has pointed a cutoff in the form of third quartile of expression, but rationale is not discussed at length.
- The authors need to explain and support all the cutoffs they used in the manuscript.
- The authors can also utilize Alternative cutpoints (median, z-score, maximally selected rank statistics or based on ROC) for optimal cutoff selection/exclusion.
- Writing and Formatting - The authors need to address all the errors that affect the flow of the manuscript (such as spacing and sentence rewriting for clarity and flow).
- The authors need to cite more recent meta-analyses/research specifically w.r.t to peritoneal metastasis. This will contextualize the importance of this research.
- Define all abbreviations at the instance of first use (for e.g., KD, mAb, HIPEC). Please ensure that it is uniformly used throughout the manuscript.
Author Response
Comments1: The authors need to strengthen the downstream signaling specifically in context of CRC.
Response1: Thank you for your valuable comment. We agree that clarifying the downstream signaling pathways regulated by CHRNB2 in the context of colorectal cancer (CRC) would enhance the mechanistic significance of our findings. In response, we have revised the manuscript to include a more detailed discussion of potential downstream effectors and signaling cascades relevant to CRC, based on previous literature.(P10, Line284-290)
Comments2: The patient cohort spans 2007–2017; please add another table for discussing clinico-pathological features/characteristics in detail.
Response2: We have submitted Supplemental Table 2 showing the association between clinicopathological factors and CHRNB2 expression. Is it necessary to provide a more detailed table? If so, could you kindly advise which specific items should be included? Thank you for your guidance.
Comments3: The manuscript has pointed a cutoff in the form of third quartile of expression, but rationale is not discussed at length.The authors need to explain and support all the cutoffs they used in the manuscript. The authors can also utilize Alternative cutpoints (median, z-score, maximally selected rank statistics or based on ROC) for optimal cutoff selection/exclusion.
Response3: We appreciate the reviewer’s suggestion regarding the use of alternative cutoff strategies. In our additional analysis, the median cutoff did not yield statistically significant differences. However, the cutoff identified using ROC curve analysis resulted in outcomes comparable to those obtained using the third quartile. Given that the third quartile is a commonly used and less arbitrary threshold in gene expression studies, we decided to retain it as our primary cutoff value. This rationale has now been included in the revised manuscript.(P7, Line228,299)
Comments4: Writing and Formatting - The authors need to address all the errors that affect the flow of the manuscript (such as spacing and sentence rewriting for clarity and flow).
Response4: We will consider revising the manuscript after further discussion with the editor.
Comments5: The authors need to cite more recent meta-analyses/research specifically w.r.t to peritoneal metastasis. This will contextualize the importance of this research.
Response5: Two recent meta-analyses regarding colorectal cancer peritoneal metastasis have been incorporated into the Introduction to reflect the latest evidence.(P3, Line55-60)
Comments6: Define all abbreviations at the instance of first use (for e.g., KD, mAb, HIPEC). Please ensure that it is uniformly used throughout the manuscript.
Response6: Thank you for pointing this out. We have carefully reviewed the entire manuscript and ensured that all abbreviations are clearly defined at their first appearance (e.g., KD for knockdown, mAb for monoclonal antibody, HIPEC for hyperthermic intraperitoneal chemotherapy). We have also confirmed that the abbreviations are used consistently and uniformly throughout the text, tables, and figure legends.

Reviewer 2 Report
Comments and Suggestions for Authors
- CRC is a heterogeneous disease, while with various consensus molecular subtypes (CMS), microsatellite status and driver mutations that influence prognosis and treatment choice. Is there any association with CHNRB2 expression with the above subtypes? In addition, stratifying patients by these molecular subtypes could also help clarify the results shown in Figure 4, particularly in relation to overall survival (OS) and disease-free survival (FDS).
- If the baseline expression difference between high- and low- expressing cell lines in Figure 1A exceeds the knockdown effect seen in Figure 1B, then directly comparing naturally high- versus low-expressing cell lines should reveal similar (or even larger) differences in migration, invasion, and proliferation.
- Are these pathways (e.g., PI3K-AKT, Wnt/β-catenin) considered part of the classical function of CHRNB2? If not, it would be much more informative if the authors could provide supporting data showing whether CHRNB2 knockdown actually affects the expression of key components in these pathways, or at least offer some evidence or analysis that suggests how CHRNB2 might be linked to their regulation.
- CHRNB2 and CHNRB2 are mixed in this manuscript.
- Is PI3K-ACT misspelling?
- Please ensure that “CHRNB2” is italicized only when referring to the gene itself.
Author Response
Comments1: CRC is a heterogeneous disease, while with various consensus molecular subtypes (CMS), microsatellite status and driver mutations that influence prognosis and treatment choice. Is there any association with CHNRB2 expression with the above subtypes? In addition, stratifying patients by these molecular subtypes could also help clarify the results shown in Figure 4, particularly in relation to overall survival (OS) and disease-free survival (DFS).
Response1: Thank you very much for this insightful comment. We fully agree that the heterogeneity of colorectal cancer—including CMS classification, microsatellite instability (MSI) status, and major driver mutations—has important implications for prognosis and therapeutic strategies.
However, as our study was based on a retrospective cohort collected between 2007 and 2017, these molecular classifications were not routinely evaluated at the time of diagnosis or treatment. Therefore, data on CMS, MSI status, and specific gene mutations were not available for our patient cohort.
We acknowledge that integrating these molecular features—especially through immunohistochemistry or transcriptomic analysis—would provide valuable insight into the biological role of CHNRB2 across CRC subtypes. We have added this limitation and future perspective to the revised Discussion section. (P10-11, Line320-322)
Commnents2: If the baseline expression difference between high- and low- expressing cell lines in Figure 1A exceeds the knockdown effect seen in Figure 1B, then directly comparing naturally high- versus low-expressing cell lines should reveal similar (or even larger) differences in migration, invasion, and proliferation.
Response2: Thank you for your insightful comment. We agree that, in principle, naturally high- versus low-expressing cell lines may show functional differences reflective of gene expression levels. However, we believe that in the context of CHRNB2, this relationship is not linear or uniform across cell lines.
Cellular functions such as proliferation, migration, and invasion are regulated not by CHRNB2 alone but by a complex network of interacting signaling pathways. In our study, LOVO cells, which exhibited relatively high CHRNB2 expression, showed significant suppression of these functions upon CHRNB2 knockdown. This suggests that in LOVO cells, CHRNB2 plays a substantial and perhaps central role in maintaining these cellular functions.
In contrast, cell lines with intrinsically low CHRNB2 expression may rely on CHRNB2-independent pathways to regulate proliferation, migration, and invasion. Therefore, low CHRNB2 expression alone does not necessarily equate to low functional activity. This could explain why direct comparisons between naturally low- and high-expressing cell lines may not replicate the magnitude of functional suppression seen in knockdown experiments.
Commnents3: Are these pathways (e.g., PI3K-AKT, Wnt/β-catenin) considered part of the classical function of CHRNB2? If not, it would be much more informative if the authors could provide supporting data showing whether CHRNB2 knockdown actually affects the expression of key components in these pathways, or at least offer some evidence or analysis that suggests how CHRNB2 might be linked to their regulation.
Response3: Thank you for your insightful comment. We agree that the Wnt/β-catenin and PI3K-AKT signaling pathways are not traditionally associated with the canonical functions of CHRNB2. In our current study, we did not experimentally assess whether CHRNB2 knockdown directly influences the expression of molecules within these pathways.
To partially address your suggestion, we conducted a correlation analysis using transcriptomic data from 53 colorectal cancer cell lines in the CCLE database. Specifically, we examined the relationship between CHRNB2 expression and the expression levels of genes involved in the Wnt and PI3K-AKT signaling pathways. Among these, only FGF3 demonstrated a statistically significant correlation with CHRNB2 expression. However, we did not observe consistent correlations with other key components of either pathway.
Thus, although CHRNB2 may be involved in modulating these pathways in certain biological contexts, our current data do not provide sufficient evidence to support a functional link between CHRNB2 and the Wnt/β-catenin or PI3K-AKT signaling pathways.
Comments4: CHRNB2 and CHNRB2 are mixed in this manuscript.
Is PI3K-ACT misspelling?
Please ensure that “CHRNB2” is italicized only when referring to the gene itself.
Comments4: We appreciate the reviewer’s valuable comments and suggestions.
We have addressed all the points raised and made the corresponding revisions in the manuscript accordingly.(P10, Line281, 282)
Reviewer 3 Report
Comments and Suggestions for Authors
This manuscript investigates the role of CHRNB2 in colorectal cancer (CRC), suggesting its involvement in promoting peritoneal dissemination and highlighting it as a potential therapeutic target. While the study is promising, several areas require clarification to ensure scientific rigor and reproducibility.
First, the authors assessed CHRNB2 expression across various CRC cell lines using qPCR and subsequently performed shRNA-mediated knockdown in a CHRNB2-high cell line. However, mRNA levels may not reflect protein expression. Western blot validation of both endogenous CHRNB2 levels and knockdown efficiency is strongly recommended to confirm specificity and functional relevance.
Figures such as 1B, 1D, and 3A lack statistical annotations. The authors should include error bars, p-values, and specify the statistical tests used to support their conclusions.
For the invasion assay, it is unclear whether cell counts were obtained from independent biological replicates or a single insert. Quantification should be based on at least three independent experiments. Besides, higher-resolution images would also improve clarity and all microscopy images should include clear scale bars.
Finally, Figure 2 appears more appropriate in the context of initial functional characterization and might be better placed alongside Figure 1. The authors are encouraged to revise the figure organization to improve the narrative flow.
Comments on the Quality of English Language
It would benefit the manuscript if the authors considered language polishing to ensure flow and clarity.
Author Response
Comments1: First, the authors assessed CHRNB2 expression across various CRC cell lines using qPCR and subsequently performed shRNA-mediated knockdown in a CHRNB2-high cell line. However, mRNA levels may not reflect protein expression. Western blot validation of both endogenous CHRNB2 levels and knockdown efficiency is strongly recommended to confirm specificity and functional relevance.
Response1: Thank you very much for this important comment. We fully agree that confirming CHRNB2 protein expression is essential to validate the mRNA findings and to ensure the specificity and functional relevance of our knockdown experiments. However, due to the short revision period of seven days, we were unfortunately unable to complete the Western blot analysis during this time. We sincerely acknowledge this as a limitation of the current revision and have added a statement in the Discussion section to clarify this point. We plan to conduct and report the protein-level validation in a follow-up study.
Comments2: Figures such as 1B, 1D, and 3A lack statistical annotations. The authors should include error bars, p-values, and specify the statistical tests used to support their conclusions.
Response2: Thank you for your suggestion. We have added error bars and P values to Figures 1B, 1D, and 3A. The statistical methods used are described in the Methods section.
Comments3: For the invasion assay, it is unclear whether cell counts were obtained from independent biological replicates or a single insert. Quantification should be based on at least three independent experiments. Besides, higher-resolution images would also improve clarity and all microscopy images should include clear scale bars.
Response3: Thank you for your valuable comment. We have quantified the invasion assay by counting the number of cells per field from four independent inserts. Error bars indicating standard deviation have been added, and statistical significance is shown with P values. A scale bar has also been included in the figure. Please note that the current image resolution is at its maximum due to the limitations of the imaging system.
Comments4: Finally, Figure 2 appears more appropriate in the context of initial functional characterization and might be better placed alongside Figure 1. The authors are encouraged to revise the figure organization to improve the narrative flow.
Response4: Thank you for your insightful suggestion. In accordance with your recommendation, we have reorganized the figure layout by moving Figure 2 to follow Figure 1, as it better aligns with the initial functional characterization. We believe this adjustment improves the overall narrative flow and logical structure of the manuscript.
Round 2
Reviewer 1 Report
Comments and Suggestions for Authors
Authors responds all comments.
Reviewer 2 Report
Comments and Suggestions for Authors
All of my concerns were addressed.